# Improved Drought Characteristics in the Pearl River Basin Based on Reconstructed GRACE Solution with Enhanced Temporal Resolution

**Linju Wang** [1,2], **Menglin Zhang** [3], **Wenjie Yin** [4], **Yi Li** [1,2,*], **Litang Hu** [5] and **Linlin Fan** [6]

1   School of Civil Engineering, Architecture and Environment, Hubei University of Technology, Wuhan 430068, China; 102110843@hbut.edu.cn
2   Key Laboratory of Intelligent Health Perception and Ecological Restoration of Rivers and Lakes, Ministry of Education, Hubei University of Technology, Wuhan 430068, China
3   Beijing Water Science and Technology Institute, Beijing 100048, China; zmlin@bwsti.com
4   Satellite Application Center for Ecology and Environment, Ministry of Ecology and Environment (MEE), Beijing 100094, China; 201531470029@mail.bnu.edu.cn
5   College of Water Sciences, Beijing Normal University, Beijing 100091, China; litanghu@bnu.edu.cn
6   Changjiang River Scientific Research Institute, Wuhan 430010, China; fanlinlin@mail.crsri.cn
*   Correspondence: liyi_bnuphd@mail.bnu.edu.cn

**Abstract:** As global warming intensifies, the damage caused by drought cannot be disregarded. Traditional drought monitoring is often carried out with monthly resolution, which fails to monitor the sub-monthly climatic event. The GRACE-based drought severity index (DSI) is a drought index based on terrestrial water storage anomalies (TWSA) observed by the gravity recovery and climate experiment (GRACE) satellite. DSI has the ability to monitor drought effectively, and it is in good consistency with other drought monitoring methods. However, the temporal resolution of DSI is limited by that of GRACE observations, so it is necessary to obtain TWSA with a higher temporal resolution to calculate DSI. We use a statistical method to reconstruct the TWSA, which adopts precipitation and temperature to obtain TWSA on a daily resolution. This statistical method needs to be combined with the time series decomposition method, and then the parameters are simulated by the Markov chain Monte Carlo (MCMC) procedure. In this study, we use this TWSA reconstruction method to obtain high-quality TWSA at daily time resolution. The correlation coefficient between CSR–TWSA and the reconstructed TWSA is 0.97, the Nash–Sutcliffe efficiency is 0.93, and the root mean square error is 16.57. The quality of the reconstructed daily TWSA is evaluated, and the DSI on a daily resolution is calculated to analyze the drought phenomenon in the Pearl River basin (PRB). The results show that the TWSA reconstructed by this method has high consistency with other daily publicly available TWSA products and TWSA provided by the Center for Space Research (CSR), which proves the feasibility of this method. The correlation between DSI based on reconstructed daily TWSA, SPI, and SPEI is greater than 0.65, which is feasible for drought monitoring. From 2003 to 2021, the DSI recorded six drought events in the PRB, and the recorded drought is more consistent with SPI-6 and SPEI-6. There was a drought event from 27 May 2011 to 12 October 2011, and this drought event had the lowest DSI minimum (minimum DSI = −1.76) recorded among the six drought events. The drought monitored by the DSI is in line with government announcements. This study provides a method to analyze drought events at a higher temporal resolution, and this method is also applicable in other areas.

**Keywords:** GRACE; terrestrial water storage anomalies; reconstruction method; drought monitoring; Pearl River basin

## 1. Introduction

As a stochastic natural catastrophe, drought has historically excessively impacted humans [1,2]. A decrease in rainfall is frequently accompanied by drought, which causes

extensive damage to the environment [3] and biological communities [4]. Being one of the main results of sensitive climate change, the frequency and severity of drought are increasing [5–7]. Therefore, a detailed analysis and description of the process of drought is needed.

The physical processes of drought are non-linear and involve feedback, and there is no single, global standard definition of drought now [8–10]. Drought is categorized into meteorological drought, hydrological drought, agricultural drought, and socioeconomic drought [8,9,11]. Quantification of drought requires drought indexes, the widely used indices include the standardized precipitation index (SPI) [12], the standardized precipitation evapotranspiration index (SPEI) [13], the standardized Runoff index (SRI) [14], the Palmer drought severity index (PDSI) [15], the self-calibrating Palmer drought severity index (cs-PDSI) [16], etc. Different drought indexes provide numerous perspectives for drought analysis [17–19]. Traditionally, the variables involved in drought indexes are based on ground-based point observations, such as hydrological and meteorological stations [20]. However, situ measurements are limited to uneven distribution, difficulty in determining spatial scale, and high cost of human and material resources. Nowadays, drought monitoring is gradually moving away from reliance on station data. The advantage of remote sensing relative to traditional ground observations is that, over a larger spatial and temporal scale, it can obtain crucial characteristics connected to drought [21,22].

The gravity recovery and climate experiment (GRACE) satellite and the GRACE follow on (GRACE-FO) mission were jointly launched by the National Aeronautics and Space Administration (NASA) and the German Aerospace Centre (DLR) to provide a method for monitoring terrestrial water storage (TWS) [23,24]. TWS is defined as the addition of all groundwater and surface water on the land, including root zone soil moisture, surface soil moisture, groundwater, and so on [25,26]. GRACE and GRACE-FO are capable of monitoring the time variable gravity field of reflection changes in mass principally caused by the Earth's water cycle with high accuracy at certain scales [27], providing valuable information for the analysis of hydrological phenomena [28]. The GRACE-based drought severity index (DSI) [29] provides a perspective on drought events based on TWS. Drought can be analyzed from multiple variables, which broadens the scope of drought monitoring. GRACE has been extensively validated and used in drought monitoring and analysis of hydrological phenomena [30–33]. Thomas et al. [34] utilized GRACE to establish a groundwater drought index for the assessment of groundwater drought in the Central Valley of California. Mohamed et al. [35] integrated GRACE, climate model outputs, and precipitation data to study groundwater variations in Chad. Sun et al. [36] derived water storage deficit based on GRACE. Wu et al. [30] used the TWSA-based total storage deficit index (TSDI) to analyze drought in five southwestern provinces of China. Sinha et al. [37] combined TWSA from GRACE with rainfall analysis to construct the combined climatologic deviation index (CCDI) to study drought in Indian river basins. The purpose of the GRACE satellite is to monitor changes in the time variable gravity field with a time interval of about 30 days [23,38]. Therefore, many scholars reconstructed it to improve its spatial resolution [39,40] or to fill in the gap time [41].

As the effects of drought continue to worsen, the short-term damage of drought cannot be disregarded [42], and it is indispensable to use more accurate time resolution to investigate the development of drought on a daily resolution, the daily evapotranspiration deficit index (DEDI) developed by Zhang et al. [43] is to explore drought on a daily resolution. The traditional DSI does not achieve monitoring droughts with a temporal resolution of shorter than one month [44,45], which cannot accurately locate the daily spatial shift of drought. The current daily TWSA provided by other institutions that can be used directly are the global land data assimilation system (GLDAS) [46,47] and ITSG-Grace2018 [48,49]. Nevertheless, certain limitations exist regarding the current availability of daily TWSA products, primarily due to discrepancies between TWSA products and GRACE TWSA in specific regions of China [50]. Additionally, the daily TWSA products are limited to infrequent updates and relatively short time series. Humphrey and Gudmundsson [51]

develop a novel approach to reconstruct TWSA on a daily resolution, utilizing daily precipitation and temperature as the driving variables. This method can not only separate human-driven and climate-driven TWSA changes [52] but also monitor mega-floods [53]. Bai et al. [54] indicate that the reconstructed TWSA from this method has higher quality. Yang et al. [55] used this method to fill the missing TWSA from July 2017 to May 2018. However, this method has not been accomplished in the actual spatial scale of the mason solution provided by the Center for Space Research (CSR), and this study uses the spatial scale of CSR to reconstruct TWSA.

The drought caused losses of more than 5.49 billion yuan in China in 2021, as reported by the "2021 Bulletin of Flood and Drought Disaster in China" [56]. The Pearl River basin (PRB), with expansive geographic coverage and intricate meteorological conditions, assumes a vital role in the provision of water to major megacities, including Guangzhou, Hong Kong, and the Pearl River Delta region [57]. The intricate dynamics of the climate system are a contributing factor to the exacerbation of drought conditions within the PRB [20]. The occurrence of recurrent and severe drought events has resulted in substantial losses within the PRB. Consequently, ensuring water security within the PRB should be prioritized, with particular emphasis placed on the monitoring of drought occurrences. Such drought monitoring plays a crucial role in predicting the onset of disasters and facilitating the formulation of effective mitigation and preventive strategies.

The utilization of GRACE is imperative for drought quantification in the PRB. Huang et al. [20] analyzed drought events in the PRB utilizing GRACE but at a monthly resolution. The occurrence of short-term extreme climate events has become more frequent. To a certain extent, a temporal resolution of a month may not suffice for a comprehensive drought analysis. In this paper, the daily TWSA of the PRB is obtained by the TWSA reconstruction method, which in turn improves the temporal resolution of the GRACE-based drought analysis in the PRB.

In this study, we reconstruct the daily TWSA and calculate the daily DSI, aiming to explore the daily drought in the PRB. The main objectives of this work are (1) to validate the quality of the daily TWSA obtained by this reconstruction method, (2) to calculate DSI with an accurate temporal resolution to daily and compare it with other drought indexes to verify the reliability of daily DSI in assessing drought events in the PRB, and (3) to calculate DSI using the daily TWSA obtained based on the reconstruction method and to study the temporal evolution and spatial distribution of drought in the PRB.

## 2. Study Area and Datasets

### 2.1. Study Area

The PRB is one of China's major basins and is located in the southern part of the country (Figure 1). The PRB covers an area of approximately $4.42 \times 10^5$ km$^2$ [57] and is higher in the west and lower in the east. There are primarily three major tributaries in the PRB: the West River, the North River, and the East River [58]. Precipitation is mainly concentrated from April to September [58,59], and hydrological drought is more severe compared with meteorological drought [60]. Situated below the population density line in China, the PRB has a high population density and robust socioeconomic development. Hence, the potential harm inflicted by drought in this region would be substantial, which makes it a matter of significant concern. Despite being a coastal region, the PRB should pay close attention to water security because of recent increases in global warming and the frequency of droughts.

The average rainfall in the Pearl River basin exhibits a trend of decreasing from the east to the west, with the eastern coastal areas receiving higher rainfall amounts and the maximum rainfall being significantly higher compared to the western areas (Figure S1). As a result, the western regions are more susceptible to droughts. This east-to-west gradient in rainfall distribution may be attributed to various climatic and geographical factors influencing the region.

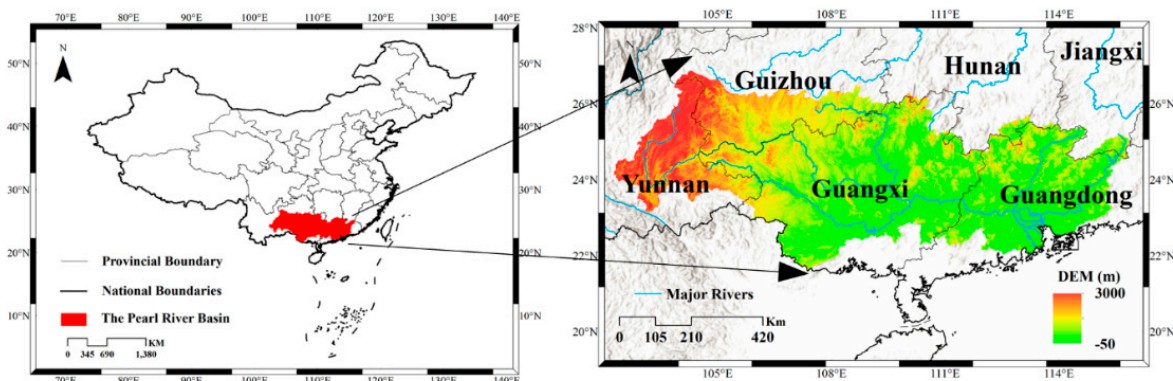

**Figure 1.** Location of the Pearl River basin and its elevation values.

*2.2. Data*

2.2.1. TWSA Products

(1)    GRACE/GRACE-FO mascon solutions: the GRACE mascon solution released by CSR (http://www2.csr.utexas.edu/grace/RL06_mascons.html (accessed on 15 June 2022)) are one of the most widely used data available today, and this study uses the GRACE/GRACE-FO RL06 Mascon Solutions (version 02) provided by CSR. In comparison to the RL05 version, the RL06 mascon solutions use a freshly established grid that can limit the leakage between land and ocean signals. The native resolution of RL06 is 1°, the shape is a square hexagon, and the file is published at 0.25° so that the coastline defined in the new RL06 mascon grid can be correctly represented [61].

(2)    Daily TWSA productions: version 2 of the GLDAS (GLDAS-2) provides optimal fields of land surface states and fluxes, which concludes TWS. The GLDAS-2.2 (https://disc.gsfc.nasa.gov/datasets/GLDAS_CLSM025_DA1_D_2.2/summary (accessed on 8 July 2022)), which assimilates TWSA (0.25° × 0.25° resolution) from GRACE, is one of the components of GLDAS-2 [46,47]. The ITSG-Grace2018 gravity field model (1° × 1° resolution, https://www2.csr.utexas.edu/grace/RL06_mascons.html (accessed on 18 December 2022)) provides Kalman smoothed daily solutions [48]. Humphrey and Gudmundsson [51] reconstruct daily TWSA using multiple precipitations and provide different products of daily TWSA (https://doi.org/10.6084/m9.figshare.7670849 (accessed on 25 December 2022)). In this study, JPL_ERA5 represents the daily TWSA reconstructed by Humphrey and Gudmundsson using the JPL-TWSA (3° × 3° resolution) and precipitation from ERA5, and JPL_MSWEP (3° × 3° resolution) represents the daily TWSA reconstructed by Humphrey and Gudmundsson using the JPL-TWSA and precipitation from MSWEP.

2.2.2. Precipitation and Temperature Data

CN05.1 (https://ccrc.iap.ac.cn/resource/detail?id=228 (accessed on 5 December 2022)) is a grid of data obtained by interpolation based on observations from 2400 Chinese meteorological stations to provide precipitation and temperature with a spatial resolution of 0.5° × 0.5° and covers the period from 1961 to 2021 [62–64]. Nie et al. [65] indicate that the TWSA reconstructed using precipitation and temperature from CN05.1 outperformed other precipitation data when using this method to reconstruct TWSA.

2.2.3. Daily Drought Index Dataset

Muliti-scale daily SPI and SPEI dataset over Mainland China: SPI, as well as SPEI, are one of the most commonly used drought indexes. To improve the temporal resolution of SPI and SPEI to identify flash droughts, Wang et al. [66,67] improved the traditional SPI and SPEI calculation methods and developed new multiscale daily SPI (https://figshare.com/articles/dataset/muliti-scale_daily_SPI_dataset_over_the_Mainland_China_from_1961-2018/14135144 (accessed on 23 December 2022)) as well as SPEI (https://figshare.com/

articles/dataset/muliti-scale_daily_SPEI_dataset_over_the_Mainland_China_from_1961-2018/12568280 (accessed on 23 December 2022)) datasets. The datasets are based on data from 484 meteorological stations in mainland China from 1961 to 2018. The new multiscale daily SPI, as well as SPEI, can effectively capture drought events in different periods and locations [68]. Table 1 shows the details of different data sets used in this study.

**Table 1.** Details of the different data sets used in this study.

| Dataset | Name | Variables | Temporal Resolution | Spatial Resolution |
| --- | --- | --- | --- | --- |
| CSR RL06 Mascon | GRACE-TWSA | TWSA | monthly | 1° × 1° **(native resolution)** |
| GLDAS | GLDAS-TWSA | TWSA | daily | 0.25° × 0.25° |
| GRACE_REC | JPL_ERA5 | TWSA | daily | 3° × 3° |
| | JPL_MSWEP | TWSA | daily | |
| ITSG-Grace2018 | ITSG-Grace2018 | TWSA | daily | 1° × 1° |
| CN05.1 | Precipitation | Precipitation | daily | 0.25° × 0.25° |
| | Temperature | Temperature | | |
| SPEI Dataset SPI Dataset | Daily SPEI Daily SPI | Daily SPEI Daily SPI | daily | Station data |

## 3. Method

In this study, we reconstructed the daily TWSA using meteorological data as well as CSR–TWSA; the meteorological data are provided by CN05.1. The reconstruction model was based on a statistical method, and the parameter of this method was calculated by the MCMC. Then, the quality of this reconstructed TWSA was analyzed in comparison with other daily TWSA products, including ITSG-Grace2018, GLDAS_TWSA, JPL_ERA5, and JPL_MSWEP, aimed to validate the feasibility of the reconstruction method. The DSI was calculated at the daily resolution using the reconstructed daily TWSA, and the calculated DSI was compared with SPI and SPEI. Finally, we calculated the drought characteristics to analyze the drought temporal distribution on a daily resolution from 2003 to 2021 and to analyze the drought special distribution in 2011 (Figure 2).

### 3.1. Recontraction Method

3.1.1. GRACE TWSA Reconstruction Method

Humphrey and Gudmundsson [51] present a statistical approach to reconstruct TWSA by assuming a linear water storage model and using precipitation and temperature forcing data. The model can be mathematically expressed as Equations (1) and (2):

$$TWSA(t) = TWSA(t-1) \cdot e^{-\frac{1}{\tau(t)}} + P(t) \tag{1}$$

$$\tau(t) = a + b \cdot T_Z(t) \tag{2}$$

where $t$ is the daily time vector, and *TWSA(t)*, *P(t)*, and *TZ(t)* represent TWSA, precipitation, and the transformation of temperature at time t, respectively. The daily TWSA obtained by providing the above method needs to be averaged into a monthly time scale corresponding to the monthly time bounds of GRACE and calibrated using the GRACE TWSA with the following monthly calibration equation:

$$anom(GRACE(t_m)) = \beta \cdot anom(TWSA(t_m)) + \varepsilon \tag{3}$$

where $\varepsilon$ denotes the error and $\beta$ denotes the calibrated scaling factor. *anom()* denotes the removal of the seasonal and trend terms from the data, and the parameters a, b, and β are

calibrated using a Markov chain Monte Carlo (MCMC) procedure. The TWSA reconstructed by this method is primarily explained by climate change [52].

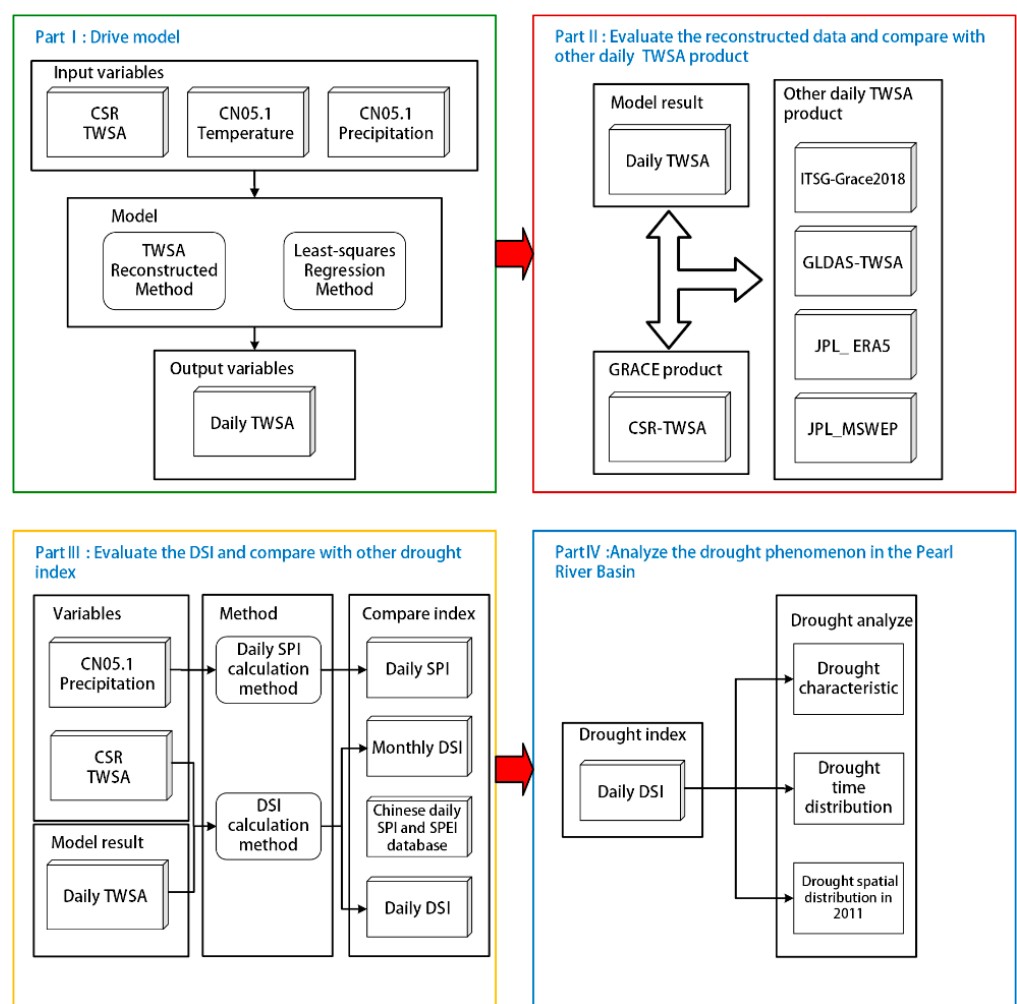

**Figure 2.** Flow chart of improved drought analysis in the Pearl River basin.

In this paper, we reconstructed the daily scale TWSA from 1 January 2003 to 31 December 2021 and adjusted the model parameters using the monthly TWSA provided by CSR from January 2003 to December 2021 as observations (missing months were not involved in parameter adjustment). Each GRACE mascon individually calibrates the above statistical model. After obtaining the parameters, we calculated the daily TWSA using Equations (1)–(3) and then interpolated the trend and seasonal terms of the original GRACE time series to obtain the daily trend and seasonal terms, and all the above terms were summed to obtain the final results.

### 3.1.2. Time Series Decomposition

The seasonal and trend terms need to be removed from TWSA in Equation (3), and the TWSA can be decomposed by the following equation [27,69]:

$$TWSA = Trend + Annual\ signal + Semi - annual\ signal + Residuals \qquad (4)$$

The trend is the linear trend of the TWSA time series. Seasonal terms consist of annual signal and semi-annual signal. Annual signal and semi-annual signal are the annual and semi-annual cycle of TWSA. The seasonal terms can be extracted by fitting sine or cosine functions; residual is the difference between TWSA and the sum of the other three

previously mentioned [54,69]. In this paper, the least-squares regression method was used to remove the linear trend as well as seasonality from the data, as in Equation (5):

$$f(t) = \underbrace{a + b \cdot (t - t_0)}_{Trend\ term} + \underbrace{\underbrace{c \cdot cos(2\pi t) + d \cdot sin(2\pi t)}_{Annual\ signal} + \underbrace{e \cdot cos(4\pi t) + f \cdot sin(4\pi t)}_{semi-annual\ signal}}_{Seasonal\ term} + \varepsilon \quad (5)$$

where $a$ is the constant term, $b$ is the trend term, $c$, $d$, $e$, and $f$ denote the seasonal term, and $\varepsilon$ denotes the residual term [70]. The parameters were obtained by the least-squares regression method. Figure 3 displays the time series as well as the trend and seasonal terms of TWSA in the PRB, which were calculated by the aforementioned method.

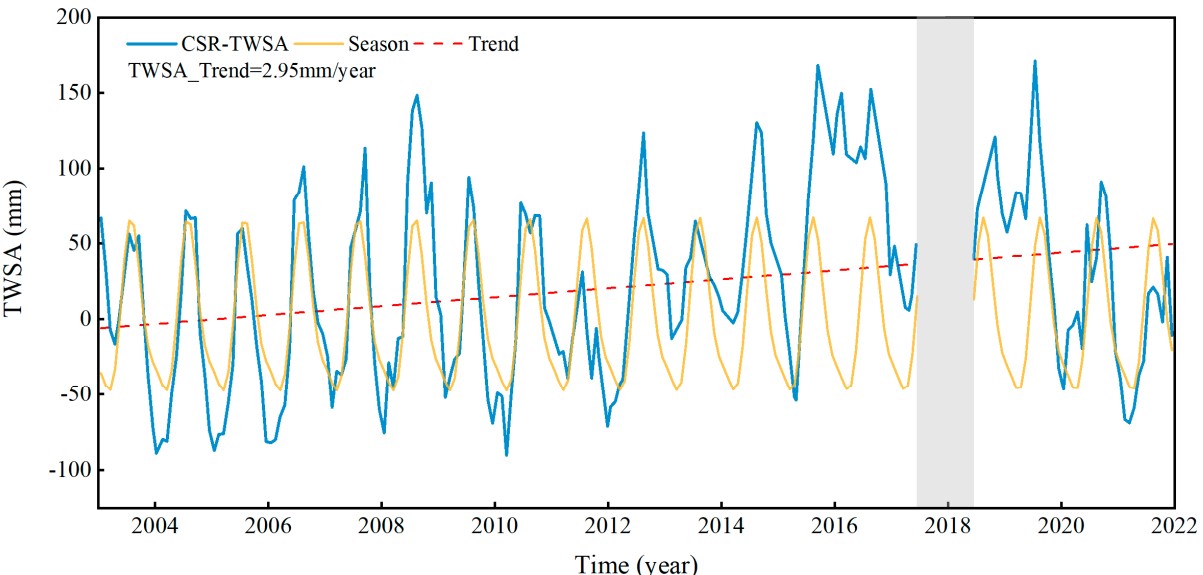

**Figure 3.** Time series of seasonal and trend terms of CSR–TWSA in the PRB.

### 3.2. Drought Index

### 3.2.1. DSI

The GRACE-based DSI was suggested by Zhao et al. [29] to identify drought phenomena. Based on this, this work extends the initial temporal resolution for DSI calculation from monthly resolution to daily resolution. Daily TWSA is used to analyze the daily drought phenomena. It can be calculated by Equation (6):

$$DSI_{i,j} = \frac{TWSA_{i,j} - \overline{TWSA_j}}{\sigma_j} \quad (6)$$

In the variable $DSI_{i,j}$ and $TWSA_{i,j}$, $i$ denotes the $i$th year, which in this paper means 19 years from 2003 to 2021, so $i = 1,2, \ldots 19$; j denotes the $j$th day of a year, $j = 1,2, \ldots 366$ in leap years and $j = 1,2, \ldots 365$ in other years; $TWSA_j$ represents the collections of TWSA on the same day in different years. For example, $TWSA_1$ is the collections of $TWSA_{1,1}$, $TWSA_{2,1}$, $TWSA_{3,1} \ldots TWSA_{19,1}$. $\overline{TWSA_j}$ and $\sigma_j$ denote the mean and standard deviation of $TWSA_j$.

The DSI was categorized into five drought categories by matching their ranking percentiles to thresholds, as recommended by the U.S. drought monitor (USDM) [29], to classify the severity of the current drought. Table 2 shows the classification of drought indices for DSI, SPI, and SPEI [29,66,67]. Generally, a drought that occurs for a period greater than or equal to three months is recorded as a drought event, and the severity of

the drought needs to be given. The formula for calculating the severity of a drought event is Equation (7) [37,44]

$$Drought\ Severity = \sum_{n}^{m} DSI(m - n \geq 90, when\ DSI \leq -0.50) \tag{7}$$

where '*n*' represents the start date of a drought event and '*m*' represents the end date of a drought event. Some academics [71] contend that the final month of a drought event is thought to be the transition between the drought period and the normal period; it should not be taken into account when determining the severity of the event. In this study, the temporal resolution of drought events was accurate to daily, and the severity of a drought event was recorded more reasonably.

**Table 2.** The range and relative categories of drought conditions for SPI, SPEI, and DSI [29,66,67].

| Category | Description | DSI | SPI | SPEI |
|----------|-------------|-----|-----|------|
| D0 | Abnormally dry | −0.50 to −0.79 | | |
| D1 | Moderate drought | −0.80 to −1.29 | −0.50 to −0.99 | −0.50 to −0.99 |
| D2 | Severe drought | −1.30 to −1.59 | −1.00 to −1.49 | −1.00 to −1.49 |
| D3 | Extreme drought | −1.60 to −1.99 | −1.50 to −199 | −1.50 to −199 |
| D4 | Exceptional drought | −2.0 or less | −2.00 or less | −2.00 or less |

3.2.2. Daily SPI

Wang et al. [67] proposed the formula for daily SPI. Firstly, the daily cumulative precipitation time series at a defined time scale (30, 90, 180 days, etc.) is calculated by the Equation (8):

$$\begin{aligned} X_{i,j}^k &= \sum_{l=31-k+j}^{30} P_{i-1,l} + \sum_{l=1}^{j} P_{i,l}, \ if\ j < k\ and \\ X_{i,j}^k &= \sum_{l=j-k+1}^{j} P_{i,l}, if\ j > k \end{aligned} \tag{8}$$

where *i* denotes the *i*th year, *j* denotes the *j*th day, and *k* (days) is the time scale. $X_{i,j}^k$ and $P_{i,j}$ denote the cumulative precipitation and the daily precipitation on day *j* of the year *i*, respectively. After that, the probability distribution of the cumulative precipitation is calculated by fitting the function of gamma probability distribution and normalizing it, and the final SPI formula is as follows. For more details, please refer to the original article [67].

$$SPI = S \frac{c_0 + W - c_1 W - c_2 W^2}{1 + d_1 W + d_2 W^2 + d_3 W^3},$$

$$W = \sqrt{\ln \frac{1}{P^2}} \begin{cases} P = 1 - F(x), \ S = -1 \ \ F(x) \leq 0.5 \\ P = 1 - P, \ S = 1 \ \ \ \ \ \ \ F(x) > 0.5 \end{cases} \tag{9}$$

*3.3. Evaluation Metrics*

In this paper, we mainly used correlation coefficient (CC) [72], root mean square error (RMSE) [73], and Nash–Sutcliffe efficiency (NSE) [74] to analyze the quality of model reconstruction results, and the three metrics are calculated as follows:

$$CC = \frac{\sum_{i=1}^{n} (X_i - \overline{X})(Y_i - \overline{Y})}{\sqrt{\sum_{i=1}^{n} (X_i - \overline{X})^2} \sqrt{\sum_{i=1}^{n} (Y_i - \overline{Y})^2}} \tag{10}$$

$$NSE = 1 - \frac{\sum_{i=1}^{n} (Y_i - X_i)^2}{\sum_{i=1}^{n} (X_i - \overline{X})^2} \tag{11}$$

$$RMSE = \sqrt{\frac{1}{n}\sum_{i=1}^{n}(Y_i - X_i)^2} \tag{12}$$

where $Y$ and $X$ represent observed and simulated values, respectively, $\overline{Y}$ and $\overline{X}$ are the average of the observed and simulated values, respectively, and n is the amount of data. The reconstructed results are more accurate the greater the NSE and CC are between the observed and simulated values. The accuracy of the model increases as the RMSE approaches zero.

## 4. Result

### 4.1. Evaluation of Reconstructed Daily TWSA

The quality of the daily TWSA derived by this method is contingent upon the accuracy of the precipitation forcing data. JPL_ERA5, JPL_MSWEP, ITSG-Grace2018, and GLDAS-TWSA were chosen to analyze the quality of the reconstructed daily TWSA in this study, with a uniform comparison from 2004 to 2016. Four ways are used to evaluate the reconstructed daily TWSA: (1) compare the reconstructed daily TWSA with the CSR–TWSA; (2) average the various daily TWSA products on a monthly scale that is consistent with the GRACE "month" and then compare them with CSR–TWSA; (3) remove the seasonal and trend terms of the various daily TWSA products, and then compare them with the reconstructed TWSA time series; and (4) determine the CC, NSE, and RMSE between the various TWSA products, to evaluate the quality of the daily TWSA reconstruction.

The reconstructed TWSA fits very well with CSR–TWSA, and the TWSA in the missing period of GRACE is also well-complemented (Figure 4). The NSE between the reconstructed TWSA's monthly mean corresponding to the GRACE time bounds and CSR–TWSA is as high as 0.92. The CSR–TWSA in the PRB shows a clear periodicity, with a significantly smaller peak of wave in 2011 than that in other years. The reconstructed data not only simulate the periodic fluctuations of the TWSA in the PRB but also the anomalous situation of a low peak of wave in 2011. The TWSA in the PRB has a clear periodicity, but in 2016, the TWSA did not show a clear trough of wave due to the high precipitation and only fluctuated in a certain value. The reconstructed data in this paper reflect the situation consistent with CSR–TWSA, indicating that the method can simulate the TWSA variation even in special periods.

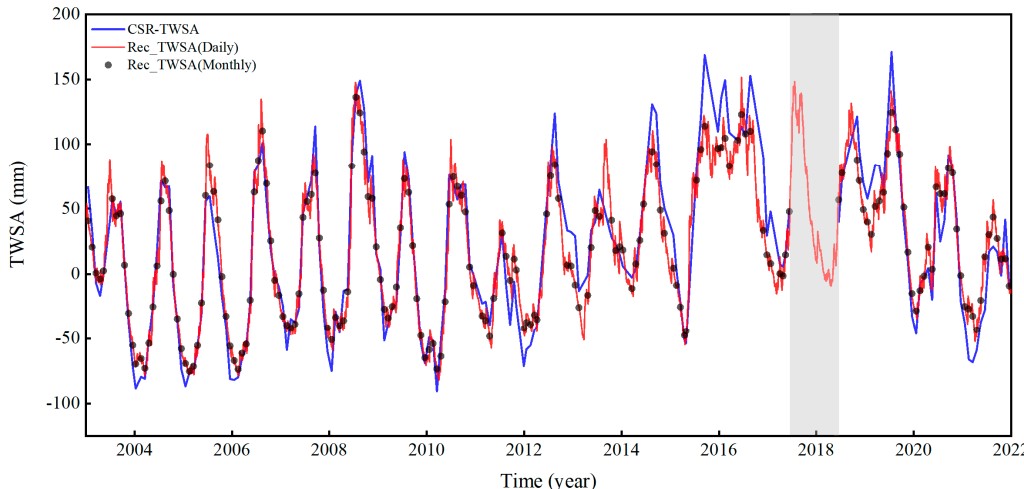

**Figure 4.** Time series of TWSA driven from CSR and reconstructed TWSA in this study. The vertical gray shading area denotes long-term missing data of GRACE.

As shown in Figure 5, there is some consistency among the different daily TWSA products, with TWSA cycles rising and falling at relatively consistent times. The monthly average of daily TWSA products is relatively smoother. Due to a large amount of data and

graphic complexity, Figure 5b is split into Figure 6a–d, the red line indicates CSR–TWSA and the blue line indicates reconstructed TWSA to examine the differences between the reconstructed TWSA and other products. The reconstructed TWSA has high consistency with other TWSA products (Figure 6). While there are some differences between the daily TWSA products in some periods and the CSR–TWSA, they are within acceptable limits. The long-term trends of all different daily TWSA products are not particularly obvious, and CSR–TWSA also shows that the trend of TWSA in the PRB is only 2.95 mm/yr (Figure 3). Humphrey and Gudmundsson [51] mentioned that long-term trends need to be carefully considered when using JPL_ERA5 and JPL_MSWEP, so the differences between these two products and CSR–TWSA may be due to long-term trends.

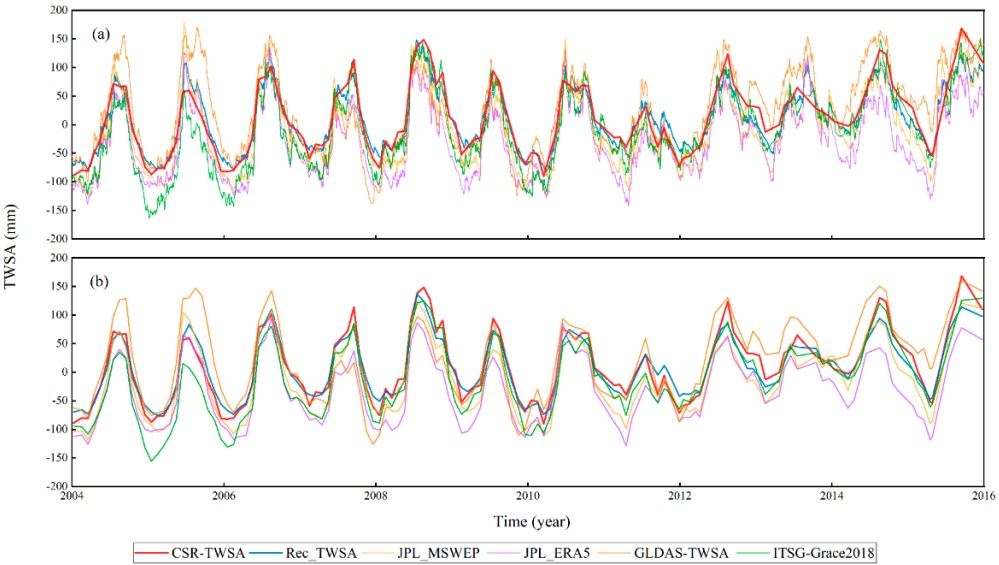

**Figure 5.** Time series of each daily TWSA product and CSR–TWSA; (**a**) CSR–TWSA and all daily TWSA products; (**b**) CSR–TWSA and each daily TWSA product averaged into the monthly scale that is consistent with the GRACE "month".

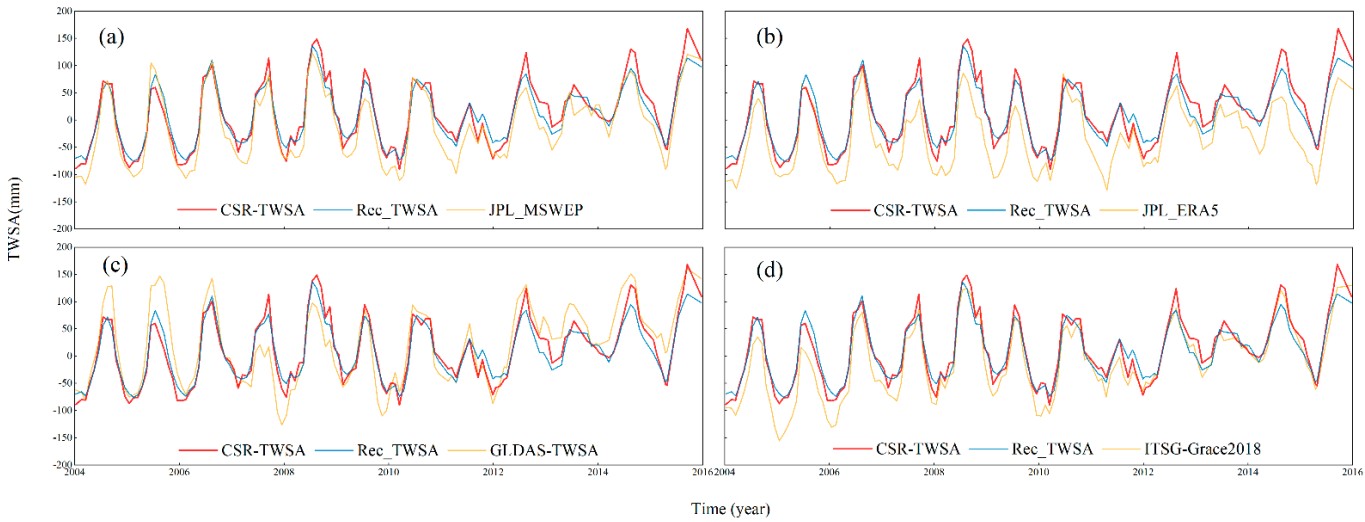

**Figure 6.** Time series of each daily TWSA product and CSR–TWSA; (**a**) time series of CSR–TWSA, reconstructed TWSA and JPL_MSWEP; (**b**) time series of CSR–TWSA, reconstructed TWSA and JPL_ERA5; (**c**) time series of CSR–TWSA, reconstructed TWSA and GLDAS-TWSA; (**d**) time series of CSR–TWSA, reconstructed TWSA and ITSG-Grace2018.

Figure 7 shows the time series of different daily TWSA products after removing the seasonal and trend terms. After removing the trend and seasonal terms, it can reflect the fluctuation of TWSA due to other conditions other than time. After removing the seasonal and trend terms, the reconstructed TWSA is still in outstanding consistency with other daily TWSA products, especially with the time series of JPL_ERA5 and JPL_MSWEP, but the difference with GLDAS-TWSA is relatively large. The discrepancy primarily manifests in the varying amplitudes during the ascent and descent of TWSA. The difference between the reconstructed TWSA and JPL_ERA5 and JPL_MSWEP is mainly determined by the quality of the precipitation [51]. Combining the results in Figures 5 and 6, in general, the reconstructed TWSA has remarkable consistency with other TWSA products in terms of time series, and the feasibility of the reconstruction method in this study is verified.

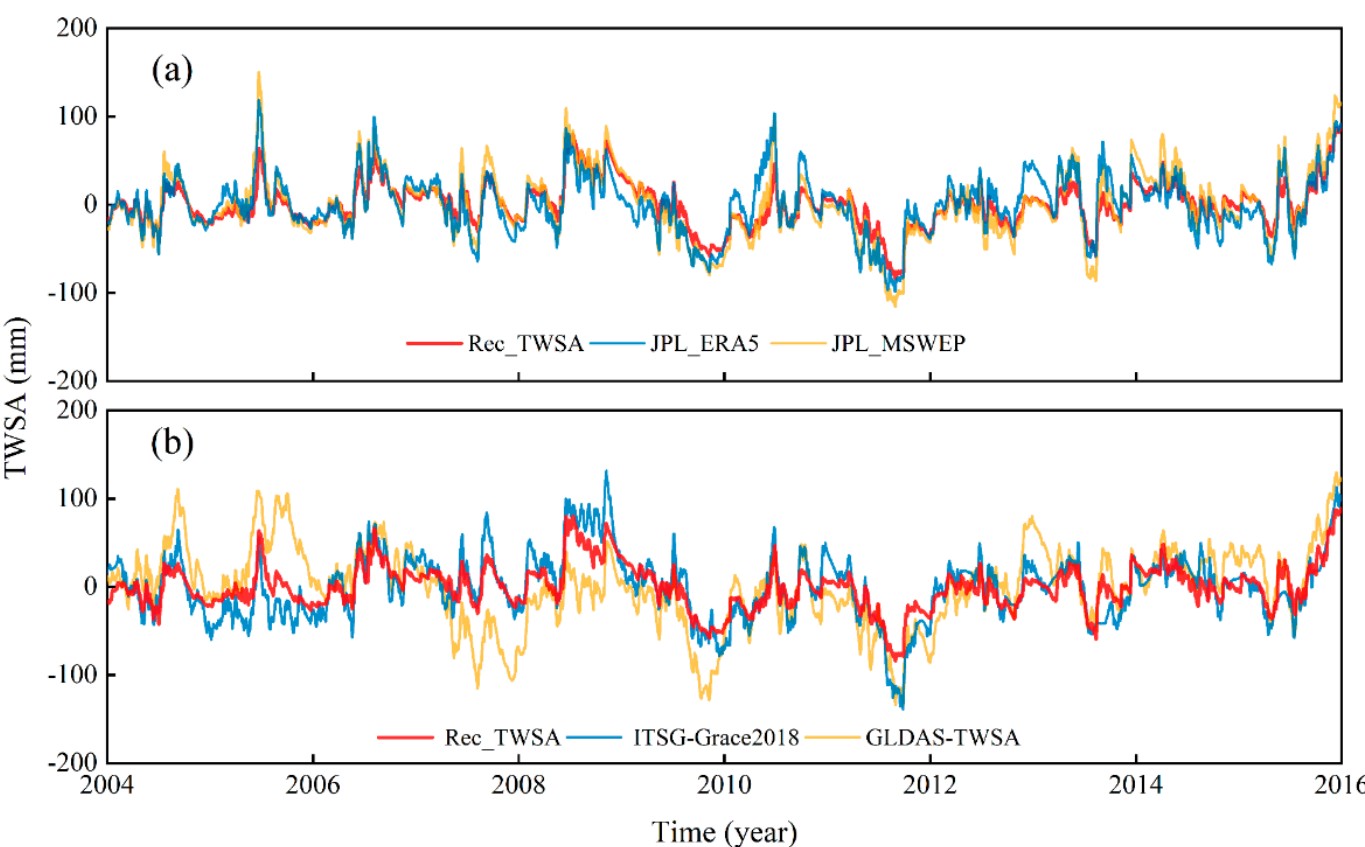

**Figure 7.** Comparison of different daily TWSA products after removing trend and seasonal terms; (**a**) time series of reconstructed TWSA in this study, JPL_ERA5, JPL_MSWEP; (**b**) time series of reconstructed TWSA in this paper, GLDAS-TWSA, ITSG-Grace2018.

All the various daily TWSA products in PRB have a high correlation (Figure 8a,b). The reconstructed TWSA exhibits correlation coefficients of up to 0.86 or higher with other daily TWSA datasets. Excluding GLDAS-TWSA, the correlation coefficients between the reconstructed TWSA and other products are above 0.94. After removing the trend and seasonal terms, the CC between each daily TWSA product is reduced, but the TWSA reconstructed in this study still maintains a high CC with other products. The CC between CSR–TWSA and the reconstructed TWSA is 0.97, the NSE is 0.93, and the RMSE is 16.57. CSR–TWSA and other daily TWSA products also have high CC (Figure S2). In summary, the reconstructed TWSA in this paper is consistent with other daily TWSA products, and the reconstructed TWSA is close to the CSR–TWSA in terms of graphical and numerical characteristics.

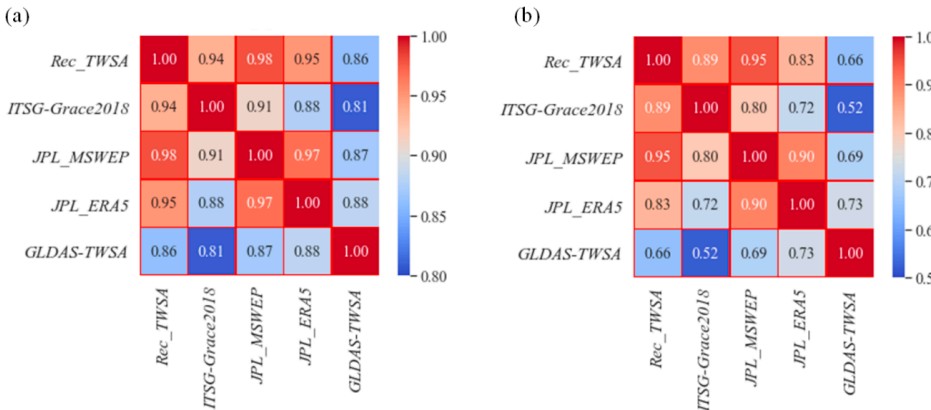

**Figure 8.** (**a**) CC between different daily TWSA products and (**b**) CC between different daily TWSA datasets with seasonal and trend terms removed.

### 4.2. Evaluation of DSI in the PRB

Figure 9a shows the time series of the daily DSI calculated based on the reconstructed daily TWSA and the monthly DSI calculated based on CSR–TWSA, where the gray area indicates the missing months of GRACE. In calculating the monthly DSI, the missing months of CSR–TWSA, except for June 2017~May 2018, are filled using cubic spline interpolation. The DSI of June 2017~May 2018 was not calculated because GRACE had long continuous missing values during this period. The time series plot exhibits strong coherence among different resolution DSI, particularly in the timing of drought occurrences. However, some periods demonstrate suboptimal agreement, primarily attributed to modeling errors during simulation. The daily DSI recorded six drought events, while the monthly DSI recorded seven. The difference is that the monthly DSI recorded a drought event from October 2007 to January 2011, while the daily DSI monitored 82 days of drought in that period, not exceeding 90 days, so the daily DSI did not record that period as a drought event. The DSI values for this period are analyzed in detail. The value of the monthly DSI in October 2007 was −0.52, but at the end of October 2007 and the first half of November 2007, the daily DSI fluctuated around −0.5, causing this period not to be considered as the time when the drought occurred. If the daily DSI had not fluctuated in early November 2007, this drought event would have exceeded 90 days and would also have been recorded.

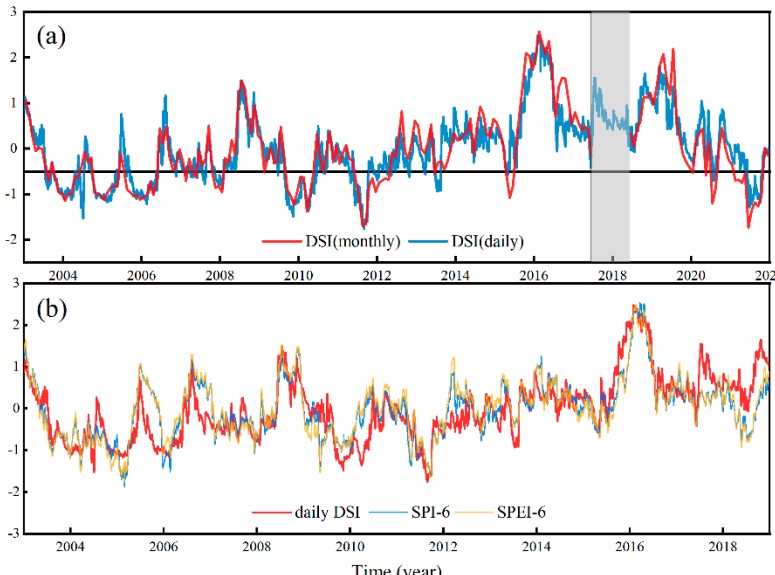

**Figure 9.** Time series of different drought indices in the PRB; (**a**) time series of monthly DSI and daily DSI; (**b**) time series of daily DSI, daily SPI-6, and daily SPEI-6.

There is a certain correlation between different drought indexes (Figure 10). The daily SPI and daily SPEI here are used from datasets provided by Wang et al. [66,67]. SPI-3 indicates daily SPI on a 3-month time scale (180-day scale). Similarly, SPI-6, SPI-12, SPEI-3, SPEI-6, and SPEI-12 follow the same pattern. The CC between DSI and other drought indexes at all time scales is above 0.65. Among them, the DSI and SPI-6 have the highest CC of 0.8. SPI and SPEI correlated over 0.95 on the same time scale. There is a certain correlation between different indices and differences, mainly due to the different perspectives of various drought indices in monitoring drought. The CC reflects the consistency of different drought indices in monitoring drought from different perspectives. DSI has the highest correlation with SPI-6 and SPEI-6, and Figure 1 shows the time series of SPI-6, SPEI-6, and DSI. Depressions are present in both SPI-6 and SPEI-6 on the dates when the DSI recorded a drought event (Figure 3), producing a wetting peak in all three indices in 2016. In summary, daily DSI is not only highly consistent with monthly DSI but also with daily SPI and daily SPEI, reflecting the reliability of daily DSI.

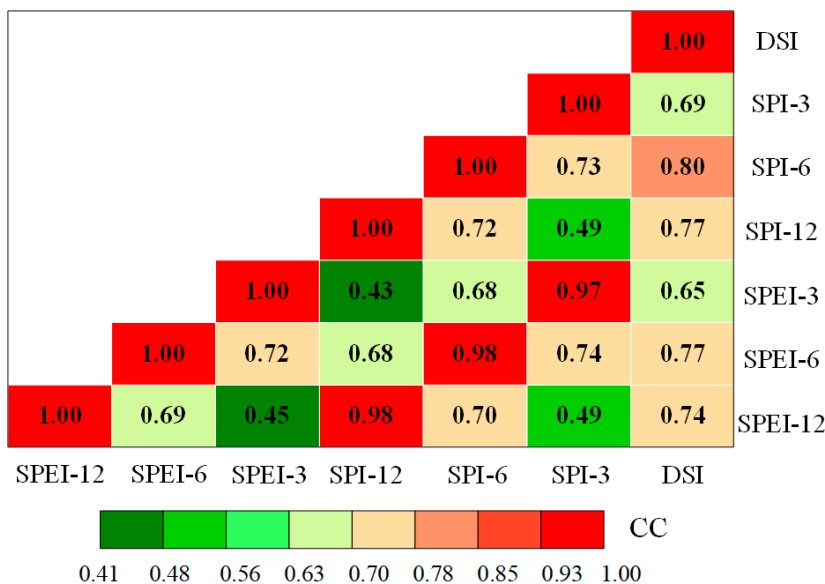

**Figure 10.** Correlation coefficients between daily DSI, daily SPI, and daily SPEI.

## 5. Discussion

### 5.1. Drought Temporal Distribution in the PRB from 2003 to 2021

Table 3 shows the characteristics of drought events recorded by DSI in PRB. Six drought events were recorded by DSI during the period from 2003 to 2021, among which the drought event occurring from 21 August 2009 to 31 May 2010 had the highest severity and the longest duration. The drought event occurring from 27 May 2011 to 12 October 2011 had the lowest DSI minimum, with a drought classification of D3. Since the next drought does not occur until 2021 after 2011, the time distribution of drought events is uneven, and droughts are more concentrated before 2011. There is a slow rise in TWSA in the PRB (Figure 2), but there is still a possibility of drought occurrence. The drought from 21 August 2009 to 31 May 2010 experienced two drought peaks, on 10 November 2009 and 31 March 2010, respectively. The DSI for the two peaks is −1.49 and −1.37, respectively, and the drought lasted for a total of 284 days, the most severe drought ever monitored by DSI. From an overall temporal perspective, the droughts that occurred in the PRB varied widely from year to year, with three drought events occurring before 2008, which were longer in duration and did not have high extreme values. The drought events that occurred afterward had higher extreme values. It is difficult to summarize the objective pattern of drought and flood occurrence in the PRB only from the temporal analysis.

**Table 3.** Statistics of DSI-identified drought events.

| ID | Duration | Duration (Days) | Total Severity | Minimum DSI (Category) | Minimum DSI Date |
|----|----------|-----------------|----------------|------------------------|------------------|
| 1 | 2003/10/1–2004/4/16 | 199 | −180.11 | −1.12 (D1) | 2004-03-17 |
| 2 | 2004/9/19–2005/5/9 | 233 | −224.98 | −1.17 (D1) | 2005-02-10 |
| 3 | 2005/9/8–2006/5/26 | 261 | −236.14 | −1.21 (D1) | 2006-02-16 |
| 4 | 2009/8/21–2010/5/31 | 284 | −298.57 | −1.49 (D2) | 2009-11-10 |
| 5 | 2011/5/27–2011/10/12 | 139 | −171.15 | −1.76 (D3) | 2011-08-31 |
| 6 | 2021/6/7–2021/10/20 | 136 | −131.12 | −1.29 (D2) | 2021-06-27 |

Six drought events were recorded by SPI-6 during the period from 2003 to 2021 (Table S1). SPI-6 here is calculated from the precipitation of CN05.1. There are similarities and differences between the drought characteristics described by DSI and those described by SPI-6. The occurrence time of each drought recorded by DSI and SPI-6 overlaps greatly, and six drought events are recorded in both drought indexes. The main reason for the difference between the two is that DSI mainly reflects hydrological drought, and SPI mainly reflects meteorological drought. There are differences in the characteristics of drought events recorded by different drought indexes.

*5.2. Spatial Distribution of Extreme Drought in the PRB in 2011*

The drought that occurred from May 27, 2011, to October 12, 2011, has the lowest minimum recorded by DSI. This study analyzes the spatial distribution of this drought event in detail. The ***"2011 Bulletin of Flood and Drought Disaster in China"*** [75] shows that the peak of the drought was in early September 2011, and the specific date of the peak of this drought recorded by DSI is 31 August 2011, with a peak value of −1.76, which is only one day different from the government recorded drought. Figure 11 shows the spatial distribution of drought monitored by DSI from 15 August 2011 to 30 September 2011. During the entire period, the PRB experienced a severe drought, with the most intense drought occurring at the tri-provincial junction of Yunnan, Guangxi, and Guizhou, and the central part of Guangdong also being significantly affected by drought. The overall drought showed a slow aggravation trend in August, and the area of D4 kept expanding. The whole drought process is spreading from west to east, reaching its peak on 31 August. The drought in the part of Guangxi, Yunnan, and Guizhou in the PRB improved slightly after 3 September but was still in a drought situation. The drought level in the whole PRB appeared to be significantly reduced on 30 September. The ***"2011 Bulletin of Flood and Drought Disaster in China"*** [75] shows that the drought was relieved by an effective precipitation process in Guangxi, Yunnan, and Guizhou in mid to late September, but the drought did not lift directly. The DSI shows that the drought in the three provinces did show a drought relief process in mid to late September, and the drought in the entire PRB was significantly reduced by the end of September 2011, which is more consistent with the description in the ***"2011 Bulletin of Flood and Drought Disaster in China"*** [75].

Both DSI and SPI-1 or SPI-6 show that there was a severe drought in August-September 2011 (Figure 12), and the drought was most severe in the tri-provincial border of Yunnan, Guangxi, and Guizhou, followed by the drought in the central part of Guangdong. SPI-1 shows that the drought in Guangxi, Yunnan, and Guizhou regions decreased from 15 August to 30 August, then briefly increased from 30 August to 5 September, and then continued to decrease until the drought largely disappeared at the end of September. Compared with the DSI, SPI-1 is lower for the drought in central Guangxi, and the drought level in central Guangxi is mostly D1 during the peak of the drought, and there are even no drought areas, so there may be an underestimation of the drought. The drought described by SPI-1 almost disappeared at the end of September is not consistent with the drought still exists in some areas in the ***"2011 Bulletin of Flood and Drought Disaster in China"*** [75], so there may be an underestimation of the drought. SPI-1 fluctuates sharply, and SPI-6 as well as DSI fluctuate slowly. The drought characteristic described by SPI-6 has been D4 in the region of Yunnan, Guangxi, and Guizhou provinces, which appears to be aggravated in

September and hardly mitigated at the end of September, which is not consistent with the description that effective precipitation makes drought mitigated in Yunnan, Guangxi, and Guizhou provinces in ***"2011 Bulletin of Flood and Drought Disaster in China"*** [75], and there may be an overestimation of drought.

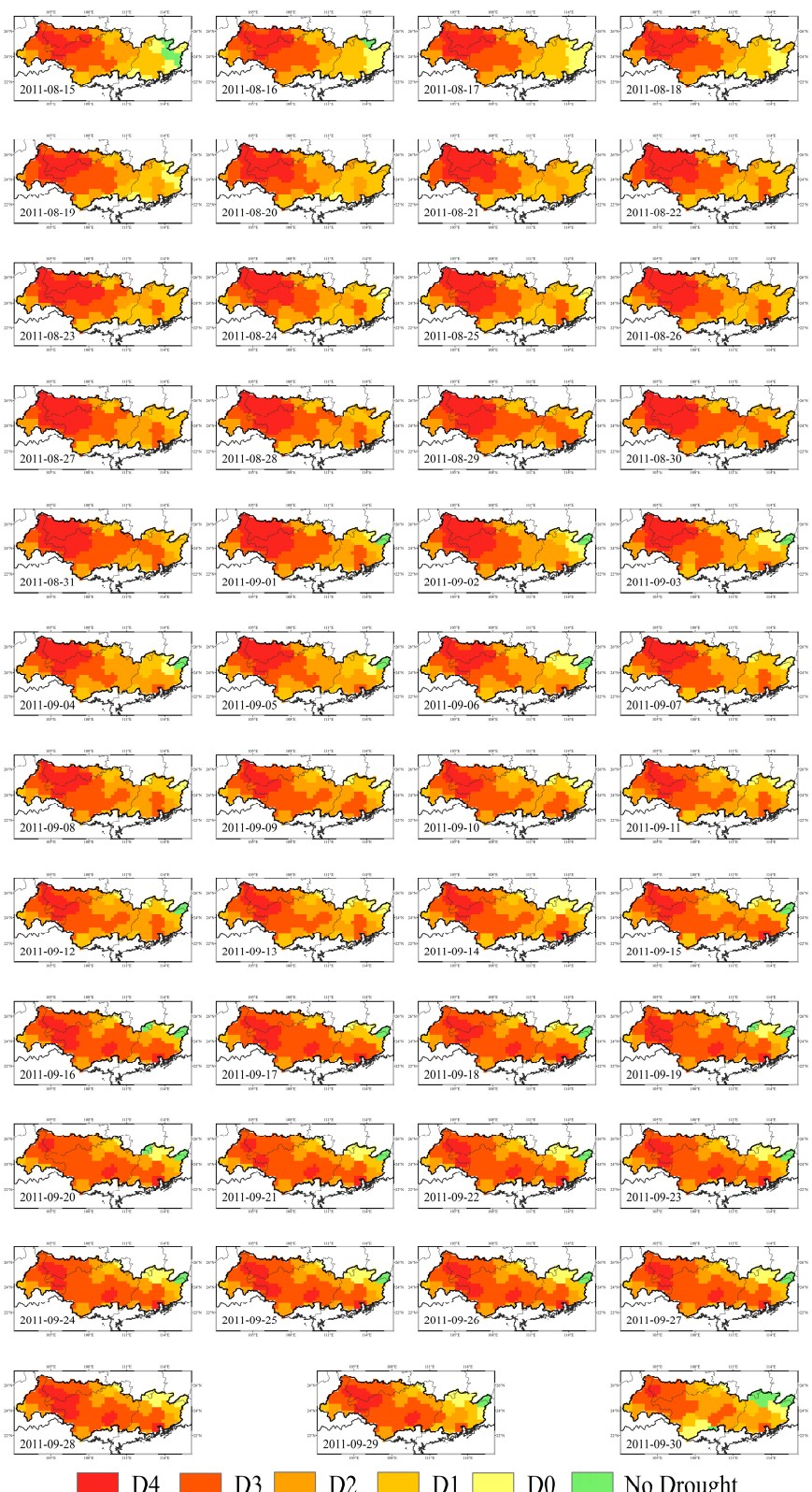

**Figure 11.** Drought distribution in the PRB monitored by DSI from 15 August to 30 September 2011.

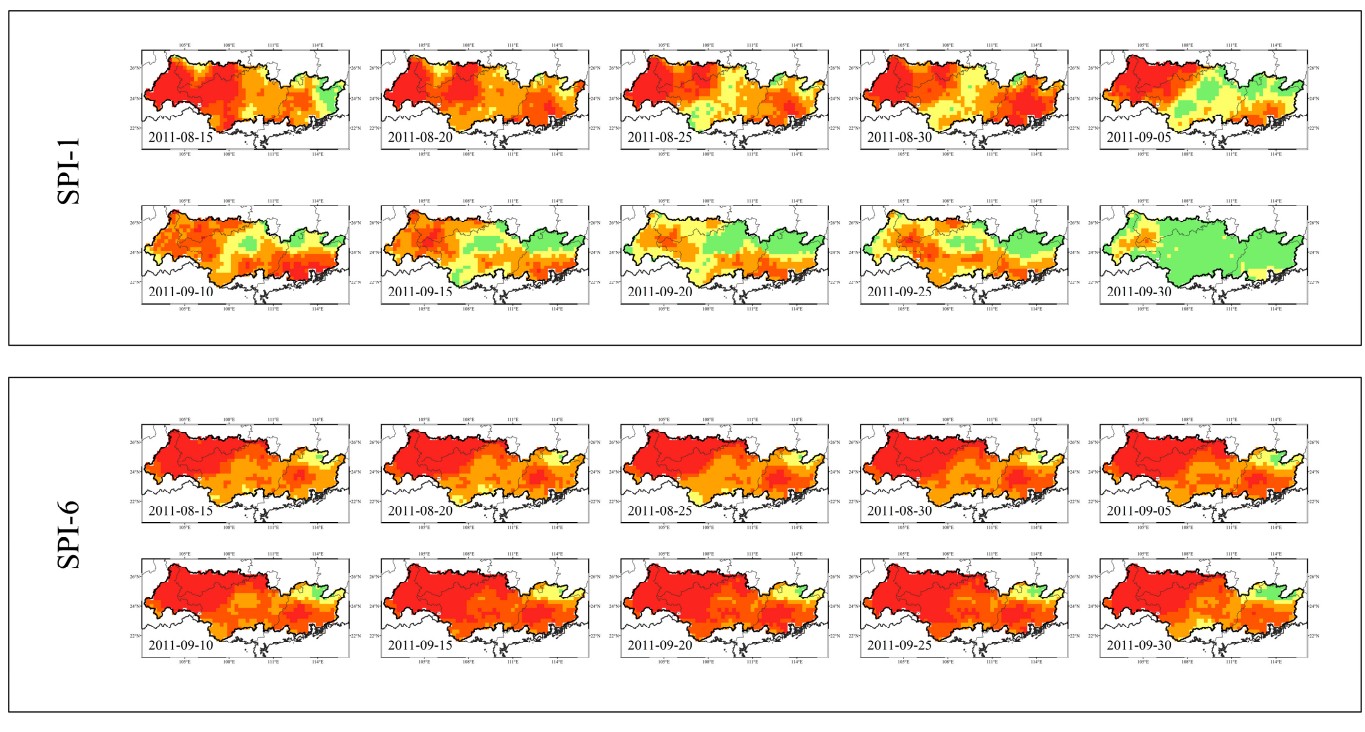

**Figure 12.** Drought distribution in the PRB monitored by SPI-1 and SPI-6 from 15 August to 30 September 2011.

The DSI is closer to the *"2011 Bulletin of Flood and Drought Disaster in China"* [75] in its description of this drought. There are differences in the description of this drought between different drought indices. The difference between SPI-1 and SPI-6 is mainly caused by the length of observation time. The main reason for the difference between DSI and SPI is that SPI studies meteorological drought, which is mainly based on the observation of precipitation, while DSI studies hydrological drought, which is based on TWSA for the analysis of drought.

## 6. Conclusions

In this study, based on the CSR–TWSA product, the reconstruction model based on a statistic method was used to reconstruct the daily TWSA and analyze the drought events in the PRB from 2003 to 2021 with the calculated DSI. We compared multiple TWSA products and compared multiple drought indexes to validate the method in this paper. The main conclusions are as follows:

(1) The quality of reconstructed TWSA using the precipitation and temperature data provided by CN05.1 is acceptable. The reconstructed TWSA is in remarkable consistency with CSR–TWSA. The NSE between the reconstructed TWSA's monthly mean corresponding to the GRACE time bounds and CSR–TWSA is as high as 0.92. The daily TWSA obtained by this method is also in noteworthy consistency with other daily TWSA products in the PRB.

(2) DSI is calculated with an improved temporal resolution to analyze more accurate drought events in the PRB. There are six drought events from 2003 to 2021 and three drought events before 2008, which have a longer duration and lower severity. The daily DSI calculated in this paper is in remarkable agreement with monthly DSI, daily SPI, and daily SPEI. The correlation coefficient between DSI and the other two is higher than 0.65. This alignment highlights the substantial significance of the DSI as a reliable metric for assessing drought conditions. The utilization of DSI with improved

temporal resolution allows the characterization of drought analysis to be studied precisely to the day, which can effectively capture the spatial evolution of drought.

(3)　In the study of drought events in the PRB in 2011, this drought event monitored by the DSI is closer to the government report than SPI-1 and SPI-6. Furthermore, the spatial distribution of drought events in all three drought indexes exhibits a relatively similar pattern, with the primary drought centers situated near the tri-provincial border of Yunnan, Guangxi, and Guizhou. From 15 August to 31 September 2011, the entirety of the PRB experienced a severe drought. Despite a brief respite during this period, drought persisted through the end of September, with a minimum DSI of 1.76 on 31 August.

This study primarily focuses on enhancing the temporal resolution in drought research. Given the temporal resolution of TWSA provided by the GRACE satellite at a monthly resolution, preceding studies have predominantly centered around monthly assessments of drought and water scarcity. This paper endeavors to augment the temporal resolution of drought research. The main limitation of this article is not to make predictions. Simulating future daily TWSA poses a challenge, primarily due to the reliance on daily precipitation as a key variable in this simulation. While simulating monthly precipitation for future periods is comparatively more feasible, accurate prediction of daily precipitation remains considerably difficult. This simulation necessitates daily precipitation data of high quality to yield a more precise TWSA estimation. Prospective research directions for this model may center on prediction. If a robust model capable of accurately predicting daily precipitation becomes available, it may be worthwhile to explore the application of simulated daily precipitation data with this model for forecasting future TWSA.

**Supplementary Materials:** The following supporting information can be downloaded at: https://www.mdpi.com/article/10.3390/rs15194849/s1, Figure S1 showing spatial distribution of mean and maximum precipitation in the Pearl River basin; Figure S2 showing the CC between different daily TWSA products and CSR–TWSA; Table S1 highlighting the characteristics of SPI-6-identified drought events.

**Author Contributions:** Conceptualization, L.W.; data curation, M.Z.; formal analysis, L.W.; funding acquisition, W.Y., L.H. and L.F.; investigation, M.Z.; methodology, L.W.; project administration, L.F.; resources, Y.L.; software, L.W.; supervision, L.H.; validation, L.W. and Y.L.; writing—original draft, L.W.; writing—review and editing, W.Y. and L.F. All authors have read and agreed to the published version of the manuscript.

**Funding:** This research was supported in part by the National Key Research and Development Project (No.2021YFC3000202), in part by the Key R&D Program of Hebei Province (21374201D), in part by the Chongqing technology innovation and application development special key project (CSTB2022TIAD-KPX0198), and in part by the National Natural Science Foundation of China (41902248).

**Data Availability Statement:** No new data were created or analyzed in this study. Data sharing is not applicable to this article.

**Acknowledgments:** The authors very appreciate the Center for Space Research for providing the GRACE mascon solution (https://www2.csr.utexas.edu/grace/RL06_mascons.html (accessed on 15 June 2022)), the Climate Change Research Center, Chinese Academy of Science (https://ccrc.iap.ac.cn/ (accessed on 5 December 2022)) for providing the CN05.1 dataset, and the NASA (http://agdisc.gsfc.nasa.gov/ (accessed on 8 July 2022)) for providing the datasets of GLDAS.

**Conflicts of Interest:** The authors declare no conflict of interest.

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
