# Peer review of "Improved Drought Characteristics in the Pearl River Basin Based on Reconstructed GRACE Solution with Enhanced Temporal Resolution"

_remotesensing, doi:10.3390/rs15194849_

Round 1
Reviewer 1 Report
The manuscript uses precipitation and temperature data to reconstruct daily TWSC data and uses it to analyze drought events in the Pearl River Basin. This research has very important scientific value and social significance. The manuscript is clear in thinking and rigorous in exposition, but there are still some defects in the details. I suggest minor revisions to publish.
L103 The author's description of the innovation points of the paper is not very clear.
L136 Not all data sources provide relevant download links and acquisition sources, which is not conducive to readers' reading and learning. Please add on.
L190 It is recommended to use the formula editor.
L247 In equation (4), m and n mean ?
L266, Source of Table 2?
L337 Can you verify whether it is caused by the quality of precipitation?
Section 4.1.1 and 4.1.2 suggest merging.
Author Response
Dear Reviewer,
We are very grateful to Reviewer for reviewing the paper so carefully. We have carefully considered these suggestions of Reviewer and made changes in our manuscript from point to point. Meanwhile, we have checked the Figures and grammar of this manuscript. We have re-edited the English language of the article to make the sentences clearer. According to the suggestions of Reviewers, we have added a Supporting Information. Thanks very much.
Please see the attachment.
Best regards,
Yi Li (Corresponding author)

Reviewer 2 Report
Dear All,
On the attached manuscript, I've posted the comments.
Best Wishes

Author Response

(The authors gave the same response as above.)

Reviewer 3 Report
I find that this manuscript has certain strengths (e.g., examination of a large dataset and comparison with GRACE estimate) but also suffers from a few key issues. A missed opportunity is to limit the study only to historical GRACE reconstruction. Unlike the geostatistical approaches, A key strength of linear water storage model, in my view, is to forecast future TWSA condition using projected precipitation from regional climate models, etc. The authors did not pursue this direction but chose instead to do a safe comparison with other datasets. English language need to be edit throughout the manuscript.
Overall, manuscript is relatively completed and can be accepted with minor efforts. My more comments are attached in the PDF.

English language need to be edit throughout the manuscript.
Author Response

(The authors gave the same response as above.)

Round 2
Reviewer 3 Report
Manuscript can be accepted without considering my further concerns.
Minor editing of English language required